# Long-Term Toxicities of Immune Checkpoint Inhibitor (ICI) in Melanoma Patients

Justin Tong [1], Adi Kartolo [2], Cynthia Yeung [3], Wilma Hopman [4] and Tara Baetz [1,*]

1 Department of Oncology, Queen's University, Kingston, ON K7L 5P9, Canada
2 Department of Oncology, McMaster University, Hamilton, ON L8V 5C2, Canada
3 Department of Medicine, Queen's University, Kingston, ON K7L 3N6, Canada
4 Department of Public Heath, Queen's University, Kingston, ON K7L 3N6, Canada
* Correspondence: tara.baetz@kingstonhsc.ca

**Abstract:** ICI therapy has greatly improved patient outcomes in melanoma, but at the cost of immune-related adverse events (irAEs). Data on the chronicity of irAEs, especially in real-world settings, are currently limited. We performed a retrospective chart review of 161 adult patients with melanoma treated with at least one cycle of ICI regimen in the adjuvant or metastatic setting: 129 patients received PD-1 inhibitor monotherapy and 32 received dual immunotherapy. Patients were grouped by duration of irAE: permanent (no complete resolution), long-term (resolution over a period ≥ 6 months), transient (resolution over a period < 6 months), or no irAEs. A total of 283 irAEs were reported in the whole patient population. Sixty-six (41.0%) patients developed permanent irAEs, fifteen (9.3%) experienced long-term irAEs as their longest-lasting toxicity, thirty-four (21.1%) developed transient irAEs only, and forty-six (28.6%) experienced no irAEs. Permanent irAEs occurred in 21 (65.6%) patients treated with dual immunotherapy and in 45 (34.9%) patients treated with monotherapy. The majority of permanent irAEs were endocrine-related (36.0%) or skin-related (32.4%). Grade 3–4 permanent irAEs occurred in 20 (12.4%) patients and included toxicities such as adrenal insufficiency, myocarditis, and myelitis. Fifty-three (32.9%) patients were still requiring treatment for long-term or permanent irAEs 6 months or more following the completion of ICI therapy, including twenty-four patients on thyroid hormone replacement and twenty-two on oral steroids. ICI treatment was temporarily interrupted for 64 (22.6%) irAEs and permanently discontinued due to irAEs in 38 patients (13.6% of irAEs, 23.6% of patients); additionally, 4 (2.5%) patients died of irAEs. Our findings show that ICI treatment in melanoma is associated with a wide range of toxicities that can be permanent and may have long-lasting impacts on patients, which should therefore be discussed when obtaining consent for treatment.

**Keywords:** oncology; immune checkpoint inhibitor; immunotherapy; melanoma

## 1. Introduction

The development of immune checkpoint inhibitor (ICI) therapy has transformed the treatment landscape for high-risk and advanced melanoma over the past decade. Novel ICI agents including ipilimumab (an anti-CTLA-4 monoclonal antibody), pembrolizumab, and nivolumab (both anti-PD-1 antibodies) are now routinely employed and have greatly improved patient outcomes [1]. While these therapies are effective in enhancing the immune response against tumour cells, they frequently also lead to immune-related adverse events (irAEs) due to the disruption of immunologic homeostasis. IrAEs involving virtually all organ systems have been described, including dermatologic, gastrointestinal, endocrine, respiratory, musculoskeletal, cardiac, renal, ocular, neurologic, and others [2]. Severity is graded on a scale of 1 to 5, with lower grades denoting minor events and higher grades corresponding to more serious events; grade 5 events are those that resulted in death. The management of irAEs requires a multi-faceted approach that may involve a combination of

supportive care, endocrine replacement, systemic corticosteroids, ICI therapy interruption, and additional immuno-modulatory therapies [3].

One area of growing research in the field of irAEs is the duration of toxicity. As clinicians accumulate more experience with ICIs, it is becoming increasingly apparent that many irAEs can have long-lasting or even permanent effects [4]. While some of these long-term toxicities are minimally bothersome to patients and easily managed, others may require prolonged courses of corticosteroids or other medications with their own side effect profiles, and/or generally have significant impacts on quality of life. It is, therefore, incumbent upon providers to discuss this potential for long-term or permanent toxicity with patients being offered ICI therapy. However, data on the prevalence of specific types of long-term toxicities to inform these discussions are currently lacking.

The present study reports data from our local experience at the Cancer Centre of Southeastern Ontario (CCSEO) with the aim of characterizing the long-term and permanent irAEs associated with adjuvant ICI treatment in patients with high-risk resected melanoma or palliative ICI treatment in patients with unresectable advanced or metastatic melanoma. We also report on the management of irAEs in these settings at our centre as well as the association with survival outcomes in patients who received single-agent ICI in the first-line palliative setting.

## 2. Materials and Methods

We conducted a retrospective chart review of 161 adult patients with melanoma treated with at least one cycle of an ICI regimen in the adjuvant or palliative setting. We obtained approval from our local research ethics board prior to study commencement. One hundred and twenty-nine patients received PD-1 inhibitor monotherapy, and thirty-two received dual immunotherapy (PD-1 inhibitor in combination with CTLA-4 monoclonal antibody). We collected baseline patient, tumour, and treatment characteristics including age, sex, Eastern Cooperative Oncology Group (ECOG) performance status, baseline autoimmune disease history, melanoma stage, BRAF mutation status, baseline neutrophil-to-lymphocyte ratio (NLR), baseline platelet-to-lymphocyte ratio (PLR), ICI regimen, and ICI treatment line.

We grouped patients by duration of irAE: permanent (no complete resolution), long-term (resolution over a period $\geq$ 6 months), transient (resolution over a period < 6 months), or no irAEs. The start date for irAE duration was determined by the date of clinic or hospital visit at which signs/symptoms of irAE were first identified. The end date for irAE duration was determined by the date of the clinic or hospital visit at which the irAE was declared completely resolved based on no further signs/symptoms and no further treatment required. Patients who experienced more than one irAE were categorized according to their longest-duration irAE. The characteristics of individual irAEs were also collected, including type, severity as graded by CTCAE v4.0 [5], treatment, time of onset, duration, and requirement for ICI interruption or discontinuation. An instance of irAE recurrence or progression to a higher grade despite treatment was considered a separate irAE. Objective response per RECIST 1.1 criteria [6] and overall survival were evaluated for those patients that were treated in the first-line palliative setting with PD-1 inhibitor monotherapy.

We conducted descriptive and univariate analyses to provide an overview of the baseline study population. Chi-square or Fisher's Exact test, if appropriate, were utilized for the univariate analyses. A logistic regression model was utilized to identify predictors of long-term or permanent irAEs. Objective response rates were analyzed using one-way analysis of variance. Overall survival was analyzed as a time-to-event endpoint using the Kaplan–Meier method with log-rank testing to calculate *p*-values. Statistical analysis was performed using Microsoft Excel in Microsoft Office 365 and IBM SPSS Statistics Version 26.

## 3. Results

Thirty-eight patients received ICI therapy in the adjuvant setting alone, and one hundred and twenty-three patients received ICI therapy in the palliative setting; 10 patients received ICI therapy in both adjuvant and palliative settings. Sixty-six (41.0%) patients developed permanent irAEs, fifteen (9.3%) experienced long-term irAEs as their longest-lasting toxicity, thirty-four (21.1%) developed transient irAEs only, and forty-six (28.6%) experienced no irAEs. Permanent irAEs occurred in 21/32 (65.6%) patients treated with dual immunotherapy and in 45/129 (34.9%) patients treated with monotherapy. The baseline characteristics of patients in each irAE duration group are reported in Table 1. At the cutoff date of analysis, the median duration of follow-up after first ICI treatment was 22.6 months.

The characteristics of the irAEs in each duration group are summarized in Table 2. A total of 283 irAEs were reported in the entire patient population: 111 (39.2%) were permanent, 34 (12.0%) were long-term, and 138 (48.8%) were transient. The majority of permanent irAES were endocrine-related (36.0%), skin-related (32.4%), or musculoskeletal (10.8%). Corticosteroids were used in the treatment of 153 (54.1%) irAEs, with high-dose corticosteroids ($\geq$1 mg/kg) used for 53 (18.7%) irAEs. ICI treatment was temporarily interrupted for 64 (22.6%) irAEs and permanently discontinued due to irAE in 38 patients (13.6% of irAEs, 23.6% of patients).

Patterns of irAE development in patients who received anti-PD-1 monotherapy are shown in Table 3. In this subgroup, permanent non-fatal irAEs with grade $\geq$3 at the height of severity occurred in 10 (7.8%) patients and included the following toxicities: dermatitis (3 patients), hypopituitarism (3 patients), vasculitis (1 patient), syndrome of inappropriate antidiuretic hormone (SIADH) (1 patient), acute interstitial nephritis (AIN) (1 patient), and brachial plexitis (1 patient). Two patients died: one patient treated with first-line palliative-intent pembrolizumab who developed fulminant liver failure, and one patient treated with adjuvant pembrolizumab who developed myocarditis. Thirty-eight (29.5%) patients were still on treatment for long-term or permanent irAEs 6 months or more following the completion of ICI therapy, including the following treatments: thyroid replacement (eighteen patients), chronic low-dose (<10 mg) prednisone (eight patients), topical steroids (twelve patients), adrenal hormone replacement (five patients), disease-modifying anti-rheumatic drugs (DMARDs) (two patients), and regular intravenous immunoglobulin (IVIG) (one patient). Some patients required more than one type of treatment for long-term or permanent irAEs. The mean time to first irAE development in this group was 25.0 months.

Patterns of irAE development in patients who received dual anti-CTLA-4/anti-PD-1 immunotherapy are shown in Table 4. In this subgroup, permanent non-fatal irAEs with grade $\geq$3 at the height of severity occurred in nine (28.1%) patients and included the following toxicities: dermatitis (two patients), pneumonitis (two patients), colitis (two patients), hypopituitarism (one patient), pancreatitis (one patient), myocarditis (one patient), myelitis (one patient), and type I diabetes mellitus (one patient). Two patients died: one patient with metastatic BRAF-mutant disease treated with second-line ipilimumab/nivolumab who developed colitis complicated by sepsis, and one patient treated with first-line ipilimumab/nivolumab who developed immune-related encephalitis. Fifteen (46.9%) patients were still on treatment for long-term or permanent irAEs 6 months or more following the completion of ICI therapy, including the following treatments: thyroid replacement (six patients), chronic low-dose prednisone (seven patients), topical steroids (one patient), adrenal hormone replacement (two patients), DMARDs (one patient), insulin (one patient), and steroid eyedrops (one patient). The mean time to first irAE development in this group was 14.8 months.

**Table 1.** Baseline patient, tumour, and treatment characteristics grouped by duration of irAE. Percentages describe proportion of patients in corresponding duration group. Abbreviations: ULN, upper limit of normal.

| | Total | Permanent irAE | Long-Term irAE | Transient irAE | No irAE | *p* |
|---|---|---|---|---|---|---|
| N | 161 | 66 | 15 | 34 | 46 | |
| Age | | | | | | 0.172 |
| <65 | 60 | 22 | 4 | 15 | 19 | |
| ≥65 | 101 | 44 | 11 | 19 | 27 | |
| Gender | | | | | | 0.949 |
| Male | 97 | 38 | 11 | 23 | 25 | |
| Female | 64 | 28 | 4 | 11 | 21 | |
| ECOG | | | | | | 0.634 |
| <2 | 137 | 58 | 12 | 32 | 35 | |
| ≥2 | 24 | 8 | 3 | 2 | 11 | |
| Baseline autoimmune history | 26 | 12 | 3 | 6 | 5 | 0.411 |
| Stage IV disease | 108 | 38 | 10 | 28 | 32 | 0.034 |
| BRAF mutant | 60 | 28 | 2 | 13 | 17 | 0.952 |
| Histology | | | | | | 0.221 |
| Cutaneous | 129 | 55 | 13 | 26 | 35 | |
| Non-cutaneous | 32 | 11 | 2 | 8 | 11 | |
| Brain metastases prior to ICI | 25 | 7 | 1 | 7 | 10 | 0.046 |
| Baseline NLR | | | | | | 0.669 |
| <5 | 127 | 52 | 13 | 26 | 36 | |
| ≥5 | 34 | 14 | 2 | 8 | 10 | |
| Baseline PLR | | | | | | 0.126 |
| <200 | 102 | 48 | 8 | 22 | 24 | |
| ≥200 | 59 | 18 | 7 | 12 | 22 | |
| Baseline LDH | | | | | | 0.335 |
| ≤ULN | 122 | 52 | 12 | 27 | 31 | |
| >ULN | 39 | 14 | 3 | 7 | 15 | |
| ICI regimen | | | | | | |
| Pembrolizumab | 106 | 32 | 13 | 23 | 38 | 0.006 |
| Nivolumab | 31 | 19 | 2 | 3 | 7 | 0.031 |
| Ipilimumab | 25 | 8 | 4 | 10 | 3 | 0.802 |
| Ipilimumab/Nivolumab | 32 | 21 | 0 | 8 | 3 | 0.053 |
| Other | 5 | 1 | 0 | 2 | 2 | 0.21 |
| Treatment intent/line | | | | | | |
| Adjuvant | 48 | 27 | 4 | 5 | 12 | 0.018 |
| Palliative | 123 | 46 | 11 | 32 | 34 | 0.07 |
| First-line | 102 | 39 | 9 | 27 | 27 | |
| Second-line | 41 | 12 | 5 | 10 | 14 | |
| Third-line | 9 | 2 | 1 | 4 | 2 | |
| Fourth-line | 1 | 1 | 0 | 0 | 0 | |

**Table 2.** Patterns and characteristics of irAE development. Percentages describe proportion of irAEs in corresponding duration group.

| | Total | Permanent irAE | Long-Term irAE | Transient irAE |
|---|---|---|---|---|
| | 283 | 111 | 34 | 138 |
| **irAE type** | | | | |
| Skin | 108 | 36 | 21 | 51 |
| Musculoskeletal | 25 | 12 | 0 | 13 |
| Endocrine | 45 | 40 | 2 | 3 |
| Neurological | 7 | 5 | 1 | 1 |
| Ocular | 8 | 0 | 1 | 7 |
| Gastrointestinal | 63 | 6 | 6 | 51 |
| Genitourinary | 2 | 2 | 0 | 0 |
| Respiratory | 20 | 6 | 2 | 12 |
| Cardiac | 2 | 2 | 0 | 0 |
| Rheumatologic, non-musculoskeletal | 3 | 2 | 1 | 0 |
| **irAE severity** | | | | |
| Grade 1–2 | 223 | 85 | 27 | 111 |
| Grade 3–4 | 56 | 22 | 7 | 27 |
| Grade 5 | 4 | 4 | 0 | 0 |
| Corticosteroid use for irAE | 153 | 56 | 16 | 81 |
| High-dose corticosteroid use ($\geq$1 mg/kg) | 53 | 22 | 6 | 25 |
| Additional immunosuppressant | 20 | 9 | 4 | 7 |
| ICI interruption/rechallenge | 64 | 7 | 7 | 50 |
| ICI permanent discontinuation | 38 | 18 | 4 | 16 |
| Subsequent irAE development | 167 | 61 | 20 | 86 |
| Same subsequent irAE type | 67 | 21 | 6 | 40 |
| Same or less subsequent irAE grade | 103 | 40 | 14 | 49 |
| Need for new long-term replacement therapy | 70 | 65 | 5 | 0 |
| Thyroid replacement | 25 | 24 | 1 | 0 |
| Maintenance corticosteroids | 22 | 22 | 0 | 0 |
| Additional immunosuppressant | 3 | 3 | 0 | 0 |
| Topical corticosteroid | 19 | 15 | 4 | 0 |
| Insulin | 1 | 1 | 0 | 0 |
| Need to increase existing replacement therapy | 4 | 4 | 0 | 0 |
| Any potential cosmetic issues (vitiligo, alopecia, poliosis) | 10 | 7 | 0 | 3 |
| Any previous transient irAE prior to permanent/long-term irAE | N/A | 37 | 11 | N/A |

Among 26 patients with baseline autoimmune disease, 21 (81%) experienced an irAE, including 11 (42.3%) who experienced a baseline disease flare, 9 (34.6%) who developed a permanent irAE, and 4 (15.4%) who developed a long-term irAE. Four patients required a long-term dose increase to an existing replacement therapy, namely thyroid replacement (three patients) or chronic prednisone (one patient).

**Table 3.** Patterns and characteristics of irAE development in subgroup of patients who received anti-PD-1 monotherapy. Percentages describe proportion of irAEs in corresponding duration group.

| | Total | Permanent irAE | Long-Term irAE | Transient irAE |
|---|---|---|---|---|
| | 204 | 80 | 32 | 92 |
| irAE type | | | | |
| Skin | 84 | 28 | 21 | 35 |
| Musculoskeletal | 20 | 10 | 0 | 10 |
| Endocrine | 36 | 31 | 2 | 3 |
| Neurological | 3 | 3 | 0 | 0 |
| Ocular | 2 | 0 | 1 | 1 |
| Gastrointestinal | 42 | 2 | 5 | 35 |
| Genitourinary | 2 | 2 | 0 | 0 |
| Respiratory | 12 | 2 | 2 | 8 |
| Cardiac | 1 | 1 | 0 | 0 |
| Rheumatologic, non-musculoskeletal | 2 | 1 | 1 | 0 |
| irAE severity | | | | |
| Grade 1–2 | 171 | 67 | 26 | 78 |
| Grade 3–4 | 31 | 11 | 6 | 14 |
| Grade 5 | 2 | 2 | 0 | 0 |
| Corticosteroid use for irAE | 97 | 35 | 14 | 48 |
| High-dose corticosteroid use (≥1 mg/kg) | 23 | 8 | 4 | 11 |
| Additional immunosuppressant | 8 | 5 | 2 | 1 |
| ICI interruption/rechallenge | 41 | 5 | 6 | 30 |
| ICI permanent discontinuation | 21 | 10 | 3 | 8 |
| Subsequent irAE development | 116 | 46 | 18 | 52 |
| Same subsequent irAE type | 56 | 18 | 6 | 32 |
| Same or less subsequent irAE grade | 72 | 29 | 12 | 31 |
| Need for new long-term replacement therapy | 47 | 45 | 2 | 0 |
| Thyroid replacement | 18 | 18 | 0 | 0 |
| Maintenance corticosteroids | 13 | 13 | 0 | 0 |
| Additional immunosuppressant | 3 | 3 | 0 | 0 |
| Topical corticosteroid | 14 | 12 | 2 | 0 |
| Insulin | 0 | 0 | 0 | 0 |
| Need to increase existing replacement therapy | 3 | 3 | 0 | 0 |
| Any potential cosmetic issues (vitiligo, alopecia, poliosis) | 7 | 6 | 0 | 1 |
| Any previous transient irAE prior to permanent/long-term irAE | N/A | 19 | 10 | N/A |

We conducted a multivariable logistic regression to identify potential predictors of long-term or permanent toxicity. Univariate analyses identified associations between the duration of toxicity and age, disease stage, baseline PLR, use of dual immunotherapy, and highest grade of irAE. The final regression model, which accounts for 18.9% of the variability with a non-significant Hosmer and Lemeshow test statistic of poor fit ($p = 0.717$), suggests that patients may be more likely to develop long-term or permanent irAEs if they are in the older age group, receive dual immunotherapy, or experienced grade ≥3 toxicity, whereas patients may be less likely to develop long-term or permanent irAEs

if they have stage IV disease at the time of first ICI therapy or had baseline PLR $\geq$ 200 (Table 5). However, of these predictors, only the presence of stage IV disease and the highest grade of toxicity were statistically significant with *p*-values < 0.05. The strongest association was seen with grade $\geq$3 toxicity, which carries a six-fold higher risk of long-term or permanent toxicity.

**Table 4.** Patterns and characteristics of irAE development in subgroup of patients who received dual anti-CTLA-4/anti-PD-1 therapy. Percentages describe proportion of irAEs in corresponding duration group.

|  | Total | Permanent irAE | Long-Term irAE | Transient irAE |
|---|---|---|---|---|
|  | 79 | 31 | 2 | 46 |
| **irAE type** |  |  |  |  |
| Skin | 24 | 8 | 0 | 16 |
| Musculoskeletal | 5 | 2 | 0 | 3 |
| Endocrine | 9 | 9 | 0 | 0 |
| Neurological | 4 | 2 | 1 | 1 |
| Ocular | 6 | 0 | 0 | 6 |
| Gastrointestinal | 21 | 4 | 1 | 16 |
| Genitourinary | 0 | 0 | 0 | 0 |
| Respiratory | 8 | 4 | 0 | 4 |
| Cardiac | 1 | 1 | 0 | 0 |
| Rheumatologic, non-musculoskeletal | 1 | 1 | 0 | 0 |
| **irAE severity** |  |  |  |  |
| Grade 1–2 | 52 | 18 | 1 | 33 |
| Grade 3–4 | 25 | 11 | 1 | 13 |
| Grade 5 | 2 | 2 | 0 | 0 |
| Corticosteroid use for irAE | 56 | 21 | 2 | 33 |
| High-dose corticosteroid use ($\geq$1 mg/kg) | 30 | 14 | 2 | 14 |
| Additional immunosuppressant | 12 | 4 | 2 | 6 |
| ICI interruption/rechallenge | 23 | 2 | 1 | 20 |
| ICI permanent discontinuation | 17 | 8 | 1 | 8 |
| Subsequent irAE development | 51 | 15 | 2 | 34 |
| Same subsequent irAE type | 11 | 3 | 0 | 8 |
| Same or less subsequent irAE grade | 31 | 11 | 2 | 18 |
| Need for new long-term replacement therapy | 19 | 18 | 0 | 1 |
| Thyroid replacement | 6 | 6 | 0 | 0 |
| Maintenance corticosteroids | 9 | 9 | 0 | 0 |
| Additional immunosuppressant | 1 | 1 | 0 | 0 |
| Topical corticosteroid | 2 | 2 | 0 | 0 |
| Insulin | 1 | 1 | 0 | 0 |
| Need to increase existing replacement therapy | 1 | 1 | 0 | 0 |
| Any potential cosmetic issues (vitiligo, alopecia, poliosis) | 3 | 1 | 0 | 2 |
| Any previous transient irAE prior to permanent/long-term irAE | N/A | 18 | 1 | N/A |

The subgroup analysis of patients who received single-agent ICI in the first-line palliative setting, shown in Figure 1, reveals that those with longer-duration irAEs had a significantly longer median overall survival (53.9 vs. 41.5 vs. 28.3 vs. 7.0 months in the permanent, long-term, transient, and no irAE categories, respectively, overall $p < 0.001$). This result was consistent in a 6-month landmark analysis performed to account for time bias (53.9 vs. 41.5 vs. 32.3 vs. 15.1 months, overall $p = 0.038$). A similar trend in objective response rate was also observed and is shown in Table 6.

**Table 5.** Multivariable logistic regression exploring predictors of long-term or permanent irAEs.

| | Sig. | Odds Ratio | 95% Confidence Interval | |
| --- | --- | --- | --- | --- |
| | | | Lower | Upper |
| Age $\geq$ 65 | 0.070 | 1.999 | 0.946 | 4.224 |
| Stage IV disease | 0.020 | 0.409 | 0.192 | 0.870 |
| Baseline PLR $\geq$ 200 | 0.101 | 0.536 | 0.255 | 1.129 |
| Dual ICI therapy | 0.245 | 1.798 | 0.669 | 4.835 |
| Highest grade of irAE $\geq$ 3 | <0.001 | 6.055 | 2.481 | 14.777 |

**Table 6.** Maximal response to ICI therapy among patients receiving anti-PD-1 monotherapy as first-line treatment for unresectable advanced or metastatic melanoma.

| | Total | Permanent irAE | Long-Term irAE | Transient irAE | No irAE | *p*-Value |
| --- | --- | --- | --- | --- | --- | --- |
| | 75 | 21 | 9 | 19 | 26 | |
| CR/PR | 31 (41.3%) | 13 (61.9%) | 5 (55.6%) | 7 (36.8%) | 6 (23.1%) | 0.042 |
| Stable/Mixed/DP | 44 (58.7%) | 8 (38.1%) | 4 (44.4%) | 12 (63.2%) | 20 (76.9%) | |

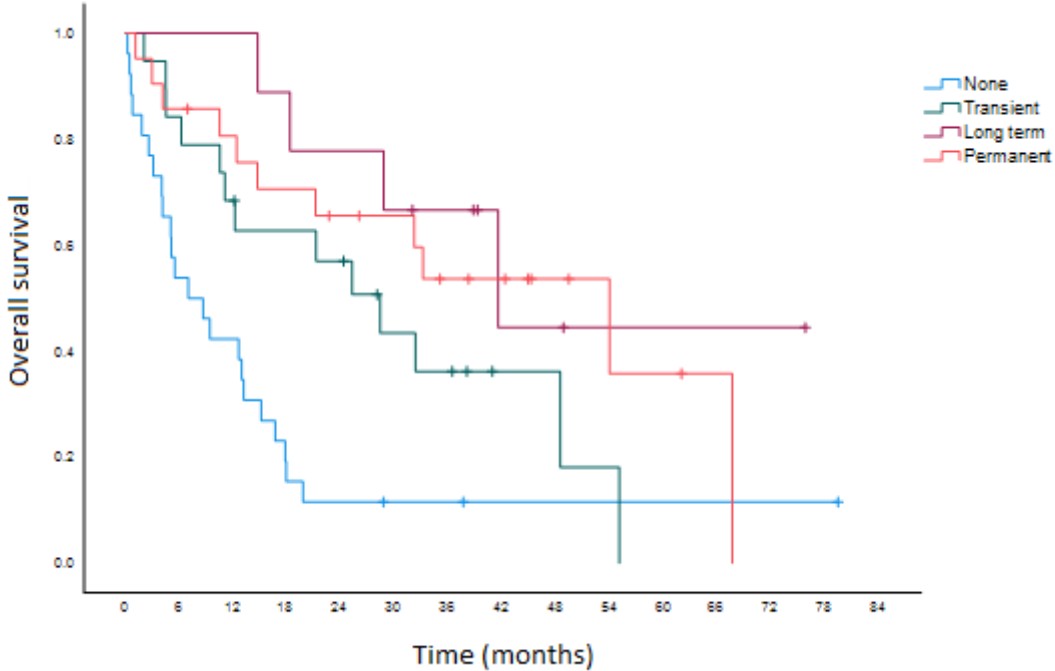

**Figure 1.** Overall survival among patients receiving anti-PD-1 monotherapy as first-line treatment for unresectable advanced or metastatic melanoma.

## 4. Discussion

We present here a retrospective observational study describing the real-world experience of an academic cancer centre with chronic toxicities secondary to ICI therapy in the treatment of adjuvant and palliative melanoma patients. Long-term or permanent irAEs, of which the majority were endocrine- or skin-related, occurred in 50.3% of patients and may be more likely in patients experiencing higher-grade irAEs. In the subgroup of patients who received single-agent anti-PD-1 as frontline therapy for advanced unresectable or metastatic disease, long-term or permanent irAEs were associated with better survival outcomes.

Treatment with an ICI regimen became the standard of care for advanced melanoma relatively recently [7–9], within the past decade, and has been utilized with increasing frequency as adjuvant therapy for resectable high-risk stage II and stage III disease over the past five years [10–13]. Consequently, there is a growing cohort of melanoma survivors who have been exposed to ICI agents, and as a corollary, there is a similarly growing need to better characterize the long-term health outcomes and chronic toxicities among these patients.

In a previous study by Johnson et al. [14], which was a retrospective chart review of 33 advanced or high-risk resected melanoma patients treated with ipilimumab who survived ≥ 2 years, 8 patients were diagnosed with hypopituitarism, with all cases requiring prolonged hormone replacement. One patient developed new-onset isolated hypothyroidism, one patient developed new-onset isolated adrenal insufficiency, one patient developed long-term neuropathic pain with possible relation to immunotherapy, and one patient had skin toxicity requiring topical corticosteroids and oral antihistamines for over one year; otherwise, all reported irAEs were transient in duration. This study's reported incidence of long-term or permanent irAEs is lower than that of our study, particularly with regard to skin toxicity; there are multiple potential reasons for this, including the fact that our study included patients who had received different ICI regimens such as combined anti-CTLA-4/anti-PD-1 therapy, which is known to be more toxic, and patients who had received multiple ICIs across multiple lines of therapy. Additionally, Johnson et al. did not explicitly define irAE resolution; as grade 1 skin toxicity carries a very low level of associated morbidity, some groups may consider improvement to grade 1 as equivalent to resolution. Purely cosmetic issues such as vitiligo, which do not typically resolve, were also not reported. As such, careful scrutiny should be exercised when comparing rates between studies.

A more recent study by Patrinely et al. [15] reported on the incidence of chronic irAEs, defined as irAEs persisting ≥ 12 weeks after therapy cessation, following adjuvant anti-PD-1 therapy for high-risk resected melanoma. This multi-centre retrospective cohort study included 387 patients and reported chronic irAEs in 167 (43.2%) patients, of which most were grade 1–2 (161 patients, 96.4%) and most were persistent at least until last available follow-up (143 patients, 85.6%). The majority of chronic irAEs were endocrine-related (54 patients, 32.3%), musculoskeletal (22 patients, 13.2%), or skin-related (19 patients, 11.4%). This ranking of affected organ systems is similar to that in our study, although the incidence of chronic skin-related irAEs in this study is lower than the incidence of permanent skin-related irAEs in our study. This is likely due, at least in part, to the fact that the Patrinely study reports irAEs as a tally of patients affected, whereas our study reports irAEs as individual events that sum to a total greater than the number of patients. Because we considered an instance of irAE recurrence or progression to a higher grade despite treatment as a separate irAE, a patient in our study experiencing a prolonged waxing-and-waning trajectory of skin toxicity that never fully resolves would translate to the irAE tally as multiple permanent irAEs of the same type. This is a potential limitation of our study and should be carefully considered when interpreting our results. Regardless, the studies are consistent in communicating the message that long-term toxicity from ICI therapy clearly warrants attention in the follow-up of melanoma patients.

The prospect of long-lasting repercussions from irAEs naturally leads to discussions around the optimal duration of ICI therapy in advanced melanoma to balance efficacy

and toxicity. It has been postulated that treatment may be safely discontinued in certain patients who have demonstrated robust responses, but the timing and identification of the best candidates for this strategy remain unclear [4]. The results of the ongoing STOP-GAP phase III study (NCT02821013) [16], which compares outcomes in metastatic melanoma patients receiving anti-PD-1 therapy intermittently versus continuously, are anticipated to partially address this issue.

Additional future areas of focus may include an analysis of the amount of healthcare resources spent on treating irAEs. As the usage of ICI therapies expands, so too will the demands for immunosuppressive therapies and irAE-related healthcare visits. Additionally, the impact of irAEs and irAE treatment on fertility and sexual potency, such as hypophysitis and long-term corticosteroids, remains yet unexplored and warrants investigation. Reproductive health is another important domain which can be affected in this population, especially as ICI therapies see increasing use in the adjuvant setting and younger patients and could be an area of focus for future studies.

The results of the current study are limited by its retrospective study design, small sample size, and inability to accurately attribute irAEs to a specific line of immunotherapy in patients who received multiple lines of immunotherapy. Particular care should be taken when interpreting the survival data, which are still affected by time bias despite the 6-month landmark analysis; of note, our sample size is too small to conduct a meaningful 12-month landmark analysis. Therefore, our findings need to be interpreted cautiously and should be validated with larger, multi-centre studies.

Nonetheless, our study provides merit in generating real-world long-term toxicity data in advanced and high-risk resected melanoma treated with ICI therapy. While ICI clinical trials typically do not include patients with known risk factors for toxicity, such as comorbid baseline autoimmune disorders, these patients do receive ICI therapy in the real world. Additionally, the risks of cosmetic or long-term toxicities, even if low grade, can have significant patient impacts that may not be captured in clinical trials.

## 5. Conclusions

The significant benefits of treatment with ICIs in melanoma are tempered by the wide range of potential toxicities that can be permanent and may have long-lasting impacts on patients. It is essential that the risk of long-term toxicities be discussed when obtaining consent for ICI therapy, and both patients and clinicians must be prepared to respond to them appropriately. The management of irAEs is becoming increasingly important as ICIs are being used more commonly in the adjuvant setting and for longer periods in the palliative setting due to the impact of survivorship.

**Author Contributions:** Conceptualization, T.B. and A.K.; methodology, A.K. and J.T.; formal analysis, J.T. and W.H.; investigation, J.T., A.K. and C.Y.; data curation, J.T.; writing—original draft preparation, J.T.; writing—review and editing, J.T., A.K. and T.B.; visualization, J.T.; supervision, T.B.; project administration, A.K. and T.B. All authors have read and agreed to the published version of the manuscript.

**Funding:** This research received no external funding.

**Institutional Review Board Statement:** The study was conducted in accordance with the Declaration of Helsinki and approved by the Institutional Review Board (or Ethics Committee) of Queen's University (protocol code ONGY-589-20, date of approval 2020/10/16).

**Informed Consent Statement:** Patient consent was waived due to the retrospective nature of this study. The research did not involve any risk to subjects, could not be carried out practically without the waiver, and the waiver did not adversely affect the rights and welfare of the subjects.

**Data Availability Statement:** The data presented in this study are available on request from the corresponding author.

**Conflicts of Interest:** The authors declare no conflict of interest.

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
