# Peer review of "Long-Term Toxicities of Immune Checkpoint Inhibitor (ICI) in Melanoma Patients"

_curroncol, doi:10.3390/curroncol29100629_

Round 1
Reviewer 1 Report
Knowledge of immunotherapy toxicity has continuously evolved, and several major cancer societies have recently published new or updated guidelines.
1. It is important to note that several cutaneous AEir can occur in the same patient and up to 9% develop multisystemic AEir, with dermatitis-pneumonitis and dermatitis-thyroiditis associations frequently observed. Skin toxicity often appears first, and it is therefore necessary to carefully monitor these toxicities. Are there any clinical data to predict this evolution?
2. The authors note, as have other authors* previously, that there is an increased risk of rare toxicities associated with sequential immunotherapies. What pathogenic explanation do they propose?
*Danlos FX, et al. Melanoma Res. 2015. Risk of a recurrence of immune toxic effects associated with anti-programmed cell death 1 antibody (anti-PD-1) therapy after discontinuation of ipilimumab monotherapy because of severe AEs.
3- Immune-related adverse events (AEs) were reported in almost 80% of patients treated with ipilimumab and high-grade events were observed in more than 20% of patients. With nivolumab, immune-related adverse events were reported in almost 60% of patients. However, regarding to pembrolizumab, serious adverse reactions were reported in 36% of patients and discontinuation of the drug due to adverse effects was reported in 9% of patients (Immunotherapy of melanoma, Rotte 2016).These data do not match those observed in this real-life study. Which explanation seems more plausible to you?
4- Considering that most of these adverse effects are immune-mediated, how do the authors explain that they are more frequent in the elderly, in whom immunosenescence may already exist?
Author Response
We thank the reviewer for their insightful comments. Please see below for our responses to individual points.
- Thank you for the observation that many patients develop multisystemic immune toxicity. In our dataset, many patients did indeed experience irAEs of multiple types. With regards to dermatitis-pneumonitis and dermatitis-thyroiditis associations specifically, at this point, there is no clinical data to support that this pattern of evolution is more likely than others. As skin toxicity is very common in patients treated with ICIs, a significant proportion of patients experiencing any particular type of irAE have previously experienced a cutaneous irAE. This is reflected in our real-world data, in which small numbers of patients have experienced both dermatitis and pneumonitis or both dermatitis and thyroiditis; however, there has been no strong signal pointing to particular patterns of evolution. Larger datasets are required to further investigate this issue.
- The increased risk of rare toxicities associated with sequential ICI therapies is interesting and is likely a reflection of the fact that these patients comprise a different population than those who receive only a single ICI therapy. This observation is potentially hypothesis-generating but we are unable to make specific conclusions based on these real-world data.
- We appreciate the reviewer drawing attention to previously reported data. Our data shows overall similar trends: irAEs were reported in 22/25 (88%) patients treated with ipilimumab, with permanent irAEs in 8/25 (32%). With nivolumab, irAEs were reported in 24/31 (77%) patients. With pembrolizumab, permanent or long-term irAEs occurred in 45/106 (42%) of patients. Grade of individual irAEs by specific ICI regimen was not tabulated, and duration of toxicity is not a surrogate for grade, but our univariate analyses did show a strong association between grade and duration of irAE. Our reported rates of any-grade toxicity are expected to be higher than those reported in clinical trials in part because many of our patients have received multiple ICI regimens across different lines of therapy. Moreover, real-world patients are a mixture of patients who are not well, tend to be older, and may have multiple tumour sites, and the approach to treatment is not standardized. Therefore, it is not surprising that there may be some differences seen between studies. Regardless, we have only described our observations.
4. We did observe a trend toward increased duration of toxicity in the elderly, but this was not a clinically significant difference. As the mechanism of irAEs themselves remains unknown, the impact of immunosenescence on toxicity from ICI therapy remains unclear and is an area that warrants further investigation.
Reviewer 2 Report
The issue of long-lasting irAEs in pts treated with ICI is relevant for patients and for clinicians and, I agree with the authors, under-reported in clinical studies, where the collection of irAEs is stopped within 6 months after the last treatment dose. Besides that, the number of health resources spent on treating irAEs is not always considered and estimated. I suggest discussing this point. There is any information regarding hospitalization needed to treat G3-4 and G5 toxicity, especially in older patients. Another important issue, not addressed in this article, is the effect on fertility and sexual potency of some toxicities (eg hypophysitis, and long-term need for steroids). This point must be investigated and discussed, especially in the adjuvant setting, and should at least be considered in the discussion, if not data could be provided
Author Response
We thank the reviewer for their insightful comments.
We agree that the amount of healthcare resources spent on treating irAEs is also an important consideration and have added a paragraph to the Discussion in which this topic is broached.
Unfortunately, hospitalization data for grade ≥3 toxicity was not collected in the current study; this could be an area of focus for future investigations. Regardless, we may direct the reviewer's attention to another study previously reported by our group*, in which we describe our centre's experience with irAEs in the emergency department. Although not specific to melanoma, this study reports real-world data of an otherwise similar population. Of 351 evaluable patients who had been treated with an ICI for any type of cancer, 129 (37%) presented to the ED at least once, and 17 of these presented with symptoms from a new irAE. Most irAEs newly diagnosed in the ED were grade ≥3, and two patients died from toxicity. Twelve patients required admission to hospital during initial presentation or follow-up, and four of these required ICU care. Evidently, there is a tangible risk of requiring an ED visit or hospitalization due to irAEs which must be clearly communicated to patients.
The reviewer's point about the effect of irAEs and irAE treatment on fertility and sexual potency is also well taken. Although we are not able to provide data relating to this issue at this time, we agree that reproductive health is another important domain which can be impacted in this population, especially as ICI therapies see increasing use in the adjuvant setting. This would certainly be an area of focus for future investigations and we have added text to the Discussion to this effect.
*Holstead, R., Kartolo, A., & Baetz, T. (2020). "Emergency Department Utilization for Patients Receiving Immune Checkpoint Inhibitors: A Retrospective Analysis of Identification and Outcomes for Those Presenting for Immune-Related Adverse Events." Curr Oncol 28(1):52-59. doi: 10.3390/curroncol28010007.